# Online Robust PCA via Stochastic Optimization

**Jiashi Feng**
ECE Department
National University of Singapore
jiashi@nus.edu.sg

**Huan Xu**
ME Department
National University of Singapore
mpexuh@nus.edu.sg

**Shuicheng Yan**
ECE Department
National University of Singapore
eleyans@nus.edu.sg

## Abstract

Robust PCA methods are typically based on batch optimization and have to load all the samples into memory during optimization. This prevents them from efficiently processing *big data*. In this paper, we develop an Online Robust PCA (OR-PCA) that processes one sample per time instance and hence its memory cost is independent of the number of samples, significantly enhancing the computation and storage efficiency. The proposed OR-PCA is based on stochastic optimization of an equivalent reformulation of the batch RPCA. Indeed, we show that OR-PCA provides a sequence of subspace estimations converging to the optimum of its batch counterpart and hence is provably robust to sparse corruption. Moreover, OR-PCA can naturally be applied for tracking dynamic subspace. Comprehensive simulations on subspace recovering and tracking demonstrate the robustness and efficiency advantages of the OR-PCA over online PCA and batch RPCA methods.

## 1   Introduction

Principal Component Analysis (PCA) [19] is arguably the most widely used method for dimensionality reduction in data analysis. However, standard PCA is brittle in the presence of outliers and corruptions [11]. Thus many techniques have been developed towards robustifying it [12, 4, 24, 25, 7]. One prominent example is the Principal Component Pursuit (PCP) method proposed in [4] that robustly finds the low-dimensional subspace through decomposing the sample matrix into a low-rank component and an overall sparse component. It is proved that both components can be recovered exactly through minimizing a weighted combination of the nuclear norm of the first term and $\ell_1$ norm of the second one. Thus the subspace estimation is robust to sparse corruptions.

However, PCP and other robust PCA methods are all implemented in a batch manner. They need to access every sample in each iteration of the optimization. Thus, robust PCA methods require memorizing all samples, in sharp contrast to standard PCA where only the covariance matrix is needed. This pitfall severely limits their scalability to *big data*, which are becoming ubiquitous now. Moreover, for an incremental samples set, when a new sample is added, the optimization procedure has to be re-implemented on all available samples. This is quite inefficient in dealing with incremental sample sets such as network detection, video analysis and abnormal events tracking.

Another pitfall of batch robust PCA methods is that they cannot handle the case where the underlying subspaces are changing gradually. For example, in the video background modeling, the background is assumed to be static across different frames for applying robust PCA [4]. Such assumption is too restrictive in practice. A more realistic situation is that the background is changed gradually along

with the camera moving, corresponding to a gradually changing subspace. Unfortunately, traditional batch RPCA methods may fail in this case.

In order to efficiently and robustly estimate the subspace of a large-scale or dynamic samples set, we propose an Online Robust PCA (OR-PCA) method. OR-PCA processes only one sample per time instance and thus is able to efficiently handle big data and dynamic sample sets, saving the memory cost and dynamically estimating the subspace of evolutional samples. We briefly explain our intuition here. The major difficulty of implementing the previous RPCA methods, such as PCP, in an online fashion is that the adopted nuclear norm tightly couples the samples and thus the samples have to be processed simultaneously. To tackle this, OR-PCA pursues the low-rank component in a different manner: using an equivalent form of the nuclear norm, OR-PCA explicitly decomposes the sample matrix into the *multiplication of the subspace basis and coefficients* plus a sparse noise component. Through such decomposition, the samples are decoupled in the optimization and can be processed separately. In particular, the optimization consists of two iterative updating components. The first one is to project the sample onto the current basis and isolate the sparse noise (explaining the outlier contamination), and the second one is to update the basis given the new sample.

Our main technical contribution is to show the above mentioned iterative optimization sheme converges to the *global optimal* solution of the original PCP formulation, thus we establish the validity of our online method. Our proof is inspired by recent results from [16], who proposed an online dictionary learning method and provided the convergence guarantee of the proposed online dictionary learning method. However, [16] can only guarantee that the solution converges to a stationary point of the optimization problem.

Besides the nice behavior on single subspace recovering, OR-PCA can also be applied for tracking time-variant subspace naturally, since it updates the subspace estimation timely after revealing one new sample. We conduct comprehensive simulations to demonstrate the advantages of OR-PCA for both subspace recovering and tracking in this work.

## 2   Related Work

The robust PCA algorithms based on nuclear norm minimization to recover low-rank matrices are now standard, since the seminal works [21, 6]. Recent works [4, 5] have taken the nuclear norm minimization approach to the decomposition of a low-rank matrix and an overall sparse matrix. Different from the setting of samples being corrupted by sparse noise, [25, 24] and [7] solve robust PCA in the case that a few samples are completely corrupted. However, all of these RPCA methods are implemented in batch manner and cannot be directly adapted to the online setup.

There are only a few pieces of work on *online robust PCA* [13, 20, 10], which we discuss below. In [13], an incremental and robust subspace learning method is proposed. The method proposes to integrate the $M$-estimation into the standard incremental PCA calculation. Specifically, each newly coming data point is re-weighted by a pre-defined influence function [11] of its residual to the current estimated subspace. However, no performance guarantee is provided in this work. In [20], a compressive sensing based recursive robust PCA algorithm is proposed. The proposed method essentially solves compressive sensing optimization over a small batch of data to update the principal components estimation instead of using a single sample, and it is not clear how to extend the method to the latter case. Recently, He *et al.* propose an incremental gradient descent method on Grassmannian manifold for solving the robust PCA problem, named GRASTA [10]. In each iteration, GRASTA uses the gradient of the updated augmented Lagrangian function after revealing a new sample to perform the gradient descent. However, no theoretic guarantee of the algorithmic convergence for GRASTA is provided in this work. Moreover, in the experiments in this work, we show that our proposed method is more robust than GRASTA to the sparse corruption and achieves higher breakdown point.

The most closely related work to ours *in technique* is [16], which proposes an online learning method for *dictionary learning* and *sparse coding*. Based on that work, [9] proposes an online *nonnegative matrix factorization* method. Both works can be seen as solving online matrix factorization problems with specific constraints (sparse or non-negative). Though OR-PCA can also be seen as a kind of matrix factorization, it is essentially different from those two works. In OR-PCA, an additive sparse noise matrix is considered along with the matrix factorization. Thus the optimization and analysis

are different from the ones in those works. In addition, benefitting from explicitly considering the noise, OR-PCA is robust to sparse contamination, which is absent in either the dictionary learning or nonnegative matrix factorization works. Most importantly, in sharp contrast to [16, 9] which shows their methods converge to a stationary point, our method is solving essentially a re-formulation of a convex optimization, and hence we can prove that the method converges to the global optimum.

After this paper was accepted, we found similar works which apply the same main idea of combining the online learning framework in [16] with the factorization formulation of nuclear norm was published in [17, 18, 23] before. However, in this work, we use different optimization from them. More specifically, our proposed algorithm needs not determine the step size or solve a Lasso subproblem.

## 3   Problem Formulation

### 3.1   Notation

We use bold letters to denote vectors. In particular, $\mathbf{x} \in \mathbb{R}^p$ denotes an authentic sample without corruption, $\mathbf{e} \in \mathbb{R}^p$ is for the noise, and $\mathbf{z} \in \mathbb{R}^p$ is for the corrupted observation $\mathbf{z} = \mathbf{x} + \mathbf{e}$. Here $p$ denotes the ambient dimension of the observed samples. Let $r$ denote the intrinsic dimension of the subspace underlying $\{\mathbf{x}_i\}_{i=1}^n$. Let $n$ denote the number of observed samples, $t$ denote the index of the sample/time instance. We use capital letters to denote matrices, *e.g.*, $Z \in \mathbb{R}^{p \times n}$ is the matrix of observed samples. Each column $\mathbf{z}_i$ of $Z$ corresponds to one sample. For an arbitrary real matrix $E$, Let $\|E\|_F$ denote its Frobenius norm, $\|E\|_{\ell_1} = \sum_{i,j} |E_{ij}|$ denote the $\ell_1$-norm of $E$ seen as a long vector in $\mathbb{R}^{p \times n}$, and $\|E\|_* = \sum_i \sigma_i(E)$ denote its nuclear norm, *i.e.*, the sum of its singular values.

### 3.2   Objective Function Formulation

Robust PCA (RPCA) aims to accurately estimate the subspace underlying the observed samples, even though the samples are corrupted by gross but sparse noise. As one of the most popular RPCA methods, the Principal Component Pursuit (PCP) method [4] proposes to solve RPCA by decomposing the observed sample matrix $Z$ into a low-rank component $X$ accounting for the low-dimensional subspace plus an overall sparse component $E$ incorporating the sparse corruption. Under mild conditions, PCP guarantees that the two components $X$ and $E$ can be exactly recovered through solving:

$$\min_{X,E} \frac{1}{2}\|Z - X - E\|_F^2 + \lambda_1\|X\|_* + \lambda_2\|E\|_1. \tag{1}$$

To solve the problem in (1), iterative optimization methods such as Accelerated Proximal Gradient (APG) [15] or Augmented Lagrangian Multiplier (ALM) [14] methods are often used. However, these optimization methods are implemented in a batch manner. In each iteration of the optimization, they need to access all samples to perform SVD. Hence a huge storage cost is incurred when solving RPCA for big data (*e.g.*, web data, large image set).

In this paper, we consider online implementation of PCP. The main difficulty is that the nuclear norm couples all the samples tightly and thus the samples cannot be considered separately as in typical online optimization problems. To overcome this difficulty, we use an equivalent form of the nuclear norm for the matrix $X$ whose rank is upper bounded by $r$, as follows [21],

$$\|X\|_* = \inf_{L \in \mathbb{R}^{p \times r}, R \in \mathbb{R}^{n \times r}} \left\{ \frac{1}{2}\|L\|_F^2 + \frac{1}{2}\|R\|_F^2 : X = LR^T \right\}.$$

Namely, the nuclear norm is re-formulated as an explicit low-rank factorization of $X$. Such nuclear norm factorization is developed in [3] and well established in recent works [22, 21]. In this decomposition, $L \in \mathbb{R}^{p \times r}$ can be seen as the basis of the low-dimensional subspace and $R \in \mathbb{R}^{n \times r}$ denotes the coefficients of the samples w.r.t. the basis. Thus, the RPCA problem (1) can be re-formulated as

$$\min_{X,L \in \mathbb{R}^{p \times r}, R \in \mathbb{R}^{n \times r}, E} \frac{1}{2}\|Z - X - E\|_F^2 + \frac{\lambda_1}{2}(\|L\|_F^2 + \|R\|_F^2) + \lambda_2\|E\|_1, \text{ s.t. } X = LR^T.$$

Substituting $X$ by $LR^T$ and removing the constraint, the above problem is equivalent to:

$$\min_{L \in \mathbb{R}^{p \times r}, R \in \mathbb{R}^{n \times r}, E} \frac{1}{2}\|Z - LR^T - E\|_F^2 + \frac{\lambda_1}{2}(\|L\|_F^2 + \|R\|_F^2) + \lambda_2\|E\|_1. \tag{2}$$

Though the reformulated objective function is not jointly convex w.r.t. the variables $L$ and $R$, we prove below that the local minima of (2) are global optimal solutions to original problem in (1). The details are given in the next section.

Given a finite set of samples $Z = [\mathbf{z}_1, \ldots, \mathbf{z}_n] \in \mathbb{R}^{p \times n}$, solving problem (2) indeed minimizes the following *empirical cost function*,

$$f_n(L) \triangleq \frac{1}{n} \sum_{i=1}^{n} \ell(\mathbf{z}_i, L) + \frac{\lambda_1}{2n} \|L\|_F^2, \tag{3}$$

where the *loss function* for each sample is defined as

$$\ell(\mathbf{z}_i, L) \triangleq \min_{\mathbf{r}, \mathbf{e}} \frac{1}{2} \|\mathbf{z}_i - L\mathbf{r} - \mathbf{e}\|_2^2 + \frac{\lambda_1}{2} \|\mathbf{r}\|_2^2 + \lambda_2 \|\mathbf{e}\|_1. \tag{4}$$

The loss function measures the representation error for the sample $\mathbf{z}$ on a fixed basis $L$, where the coefficients on the basis $\mathbf{r}$ and the sparse noise $\mathbf{e}$ associated with each sample are optimized to minimize the loss. In the stochastic optimization, one is usually interested in the minimization of the *expected cost* overall all the samples [16],

$$f(L) \triangleq \mathbb{E}_{\mathbf{z}}[\ell(\mathbf{z}, L)] = \lim_{n \to \infty} f_n(L), \tag{5}$$

where the expectation is taken w.r.t. the distribution of the samples $\mathbf{z}$. In this work, we first establish a surrogate function for this expected cost and then optimize the surrogate function for obtaining the subspace estimation in an online fashion.

## 4 Stochastic Optimization Algorithm for OR-PCA

We now present our Online Robust PCA (OR-PCA) algorithm. The main idea is to develop a stochastic optimization algorithm to minimize the empirical cost function (3), which *processes one sample per time instance* in an online manner. The coefficients $\mathbf{r}$, noise $\mathbf{e}$ and basis $L$ are optimized in an alternative manner. In the $t$-th time instance, we obtain the estimation of the basis $L_t$ through minimizing the cumulative loss w.r.t. the previously estimated coefficients $\{\mathbf{r}_i\}_{i=1}^{t}$ and sparse noise $\{\mathbf{e}_i\}_{i=1}^{t}$. The objective function for updating the basis $L_t$ is defined as,

$$g_t(L) \triangleq \frac{1}{t} \sum_{i=1}^{t} \left( \frac{1}{2} \|\mathbf{z}_i - L\mathbf{r}_i - \mathbf{e}_i\|_2^2 + \frac{\lambda_1}{2} \|\mathbf{r}_i\|_2^2 + \lambda_2 \|\mathbf{e}_i\|_1 \right) + \frac{\lambda_1}{2t} \|L\|_F^2. \tag{6}$$

This is a surrogate function of the empirical cost function $f_t(L)$ defined in (3), i.e., it provides an upper bound for $f_t(L)$: $g_t(L) \geq f_t(L)$.

The proposed algorithm is summarized in Algorithm 1. Here, the subproblem in (7) involves solving a small-size convex optimization problem, which can be solved efficiently by the off-the-shelf solver (see the supplementary material). To update the basis matrix $L$, we adopt the block-coordinate descent with warm restarts [2]. In particular, each column of the basis $L$ is updated individually while fixing the other columns.

The following theorem is the main theoretic result of the paper, which states that the solution from Algorithm 1 will converge to the optimal solution of the batch optimization. Thus, the proposed OR-PCA converges to the correct low-dimensional subspace even in the presence of sparse noise, as long as the batch version – PCP – works.

**Theorem 1.** *Assume the observations are always bounded. Given the rank of the optimal solution to (5) is provided as $r$, and the solution $L_t \in \mathbb{R}^{p \times r}$ provided by Algorithm 1 is full rank, then $L_t$ converges to the optimal solution of (5) asymptotically.*

Note that the assumption that observations are bounded is quite natural for the realistic data (such as images, videos). We find in the experiments that the final solution $L_t$ is always full rank. A standard stochastic gradient descent method may further enhance the computational efficiency, compared with the used method here. We leave the investigation for future research.

---

**Algorithm 1** Stochastic Optimization for OR-PCA

---

**Input:** $\{\mathbf{z}_1, \ldots, \mathbf{z}_T\}$ (observed data which are revealed sequentially), $\lambda_1, \lambda_2 \in \mathbb{R}$ (regularization parameters), $L_0 \in \mathbb{R}^{p \times r}$, $\mathbf{r}_0 \in \mathbb{R}^r$, $\mathbf{e}_0 \in \mathbb{R}^p$ (initial solutions), $T$ (number of iterations).
   **for** $t = 1$ to $T$ **do**
      1) Reveal the sample $\mathbf{z}_t$.
      2) Project the new sample:

$$\{\mathbf{r}_t, \mathbf{e}_t\} = \arg\min \frac{1}{2} \|\mathbf{z}_t - L_{t-1}\mathbf{r} - \mathbf{e}\|_2^2 + \frac{\lambda_1}{2}\|\mathbf{r}\|_2^2 + \lambda_2\|\mathbf{e}\|_1. \tag{7}$$

   3) $A_t \leftarrow A_{t-1} + \mathbf{r}_t\mathbf{r}_t^T$, $B_t \leftarrow B_{t-1} + (\mathbf{z}_t - \mathbf{e}_t)\mathbf{r}_t^T$.
   4) Compute $L_t$ with $L_{t-1}$ as warm restart using Algorithm 2:

$$L_t \triangleq \arg\min \frac{1}{2}\mathrm{Tr}\left[L^T\left(A_t + \lambda_1 I\right)L\right] - \mathrm{Tr}(L^T B_t). \tag{8}$$

   **end for**
   **Return** $X_T = L_T R_T^T$ (low-rank data matrix), $E_T$ (sparse noise matrix).

---

---

**Algorithm 2** The Basis Update

---

**Input:** $L = [\mathbf{l}_1, \ldots, \mathbf{l}_r] \in \mathbb{R}^{p \times r}$, $A = [\mathbf{a}_1, \ldots, \mathbf{a}_r] \in \mathbb{R}^{r \times r}$, and $B = [\mathbf{b}_1, \ldots, \mathbf{b}_r] \in \mathbb{R}^{p \times r}$.
   $\tilde{A} \leftarrow A + \lambda_1 I$.
   **for** $j = 1$ to $r$ **do**

$$\mathbf{l}_j \leftarrow \frac{1}{\tilde{A}_{j,j}}(\mathbf{b}_j - L\tilde{\mathbf{a}}_j) + \mathbf{l}_j. \tag{9}$$

   **end for**
   **Return** $L$.

---

## 5 Proof Sketch

In this section we sketch the proof of Theorem 1. The details are deferred to the supplementary material due to space limit.

The proof of Theorem 1 proceeds in the following four steps: (I) we first prove that the surrogate function $g_t(L_t)$ converges almost surely; (II) we then prove that the solution difference behaves as $\|L_t - L_{t-1}\|_F = O(1/t)$; (III) based on (II) we show that $f(L_t) - g_t(L_t) \to 0$ almost surely, and the gradient of $f$ vanishes at the solution $L_t$ when $t \to \infty$; (IV) finally we prove that $L_t$ actually converges to the optimum solution of the problem (5).

**Theorem 2** (Convergence of the surrogate function $g_t$). *Let $g_t$ denote the surrogate function defined in (6). Then, $g_t(L_t)$ converges almost surely when the solution $L_t$ is given by Algorithm 1.*

We prove Theorem 2, i.e., the convergence of the stochastic positive process $g_t(L_t) > 0$, by showing that it is a quasi-martingale. We first show that the summation of the positive difference of $g_t(L_t)$ is bounded utilizing the fact that $g_t(L_t)$ upper bounds the empirical cost $f_t(L_t)$ and the loss function $\ell(\mathbf{z}_t, L_t)$ is Lipschitz. These imply that $g_t(L_t)$ is a quasi-martingale. Applying the lemma from [8] about the convergence of quasi-martingale, we conclude that $g_t(L_t)$ converges.

Next, we show the difference of the two successive solutions converges to $0$ as $t$ goes to infinity.

**Theorem 3** (Difference of the solution $L_t$). *For the two successive solutions obtained from Algorithm 1, we have*
$$\|L_{t+1} - L_t\|_F = O(1/t) \ \ a.s.$$

To prove the above result, we first show that the function $g_t(L)$ is strictly convex. This holds since the regularization component $\lambda_1\|L\|_F^2$ naturally guarantees that the eigenvalues of the Hessian matrix are bounded away from zero. Notice that this is essentially different from [16], where one has to assume that the smallest eigenvalue of the Hessian matrix is lower bounded. Then we further show

that variation of the function $g_t(L)$, $g_t(L_t) - g_{t+1}(L_t)$, is Lipschitz if using the updating rule shown in Algorithm 2. Combining these two properties establishes Theorem 3.

In the third step, we show that the expected cost function $f(L_t)$ is a smooth one, and the difference $f(L_t) - g_t(L_t)$ goes to zero when $t \to \infty$. In order for showing the regularity of the function $f(L_t)$, we first provide the following optimality condition of the loss function $\ell(L_t)$.

**Lemma 1** (Optimality conditions of Problem (4)). $\mathbf{r}^\star \in \mathbb{R}^r$ and $\mathbf{e}^\star \in \mathbb{R}^p$ is a solution of Problem (4) if and only if

$$C_\Lambda(\mathbf{z}_\Lambda - \mathbf{e}_\Lambda^\star) = \lambda_2 \text{sign}(\mathbf{e}_\Lambda^\star),$$
$$|C_{\Lambda^c}(\mathbf{z}_{\Lambda^c} - \mathbf{e}_{\Lambda^c}^\star)| \le \lambda_2, \; otherwise,$$
$$\mathbf{r}^\star = (L^T L + \lambda_1 I)^{-1} L^T (\mathbf{z} - \mathbf{e}^\star),$$

where $C = I - L(L^T L + \lambda_1 I)^{-1} L^T$ and $C_\Lambda$ denotes the columns of matrix $C$ indexed by $\Lambda = \{j | \mathbf{e}^\star[j] \ne 0\}$ and $\Lambda^c$ denotes the complementary set of $\Lambda$. Moreover, the optimal solution is unique.

Based on the above lemma, we can prove that the solution $\mathbf{r}^\star$ and $\mathbf{e}^\star$ are Lipschitz w.r.t. the basis $L$. Then, we can obtain the following results about the regularity of the expected cost function $f$.

**Lemma 2.** *Assume the observations $\mathbf{z}$ are always bounded. Define*

$$\{\mathbf{r}^\star, \mathbf{e}^\star\} = \arg\min_{\mathbf{r}, \mathbf{e}} \frac{1}{2} \|\mathbf{z} - L\mathbf{r} - \mathbf{e}\|_2^2 + \frac{\lambda_1}{2} \|\mathbf{r}\|_2^2 + \lambda_2 \|\mathbf{e}\|_1.$$

*Then, 1) the function $\ell$ defined in (4) is continuously differentiable and*

$$\nabla_L \ell(\mathbf{z}, L) = (L\mathbf{r}^\star + \mathbf{e}^\star - \mathbf{z})\mathbf{r}^{\star T};$$

*2) $\nabla f(L) = \mathbb{E}_\mathbf{z}[\nabla_L \ell(\mathbf{z}, L)]$; and 3)$\nabla f(L)$ is Lipschitz.*

Equipped with the above regularities of the expected cost function $f$, we can prove the convergence of $f$, as stated in the following theorem.

**Theorem 4** (Convergence of $f$). *Let $g_t$ denote the surrogate function defined in (2). Then, 1) $f(L_t) - g_t(L_t)$ converges almost surely to 0; and 2) $f(L_t)$ converges almost surely, when the solution $L_t$ is given by Algorithm 1.*

Following the techniques developed in [16], we can show the solution obtained from Algorithm 1, $L_\infty$, satisfies the first order optimality condition for minimizing the expected cost $f(L)$. Thus the OR-PCA algorithm provides a solution converging to a *stationary point* of the expected loss.

**Theorem 5.** *The first order optimal condition for minimizing the objective function in (5) is satisfied by $L_t$, the solution provided by Algorithm 1, when $t$ tends to infinity.*

Finally, to complete the proof, we establish the following result stating that any full-rank $L$ that satisfies the first order condition is the *global optimal solution*.

**Theorem 6.** *When the solution $L$ satisfies the first order condition for minimizing the objective function in (5), the obtained solution $L$ is the optimal solution of the problem (5) if $L$ is full rank.*

Combining Theorem 5 and Theorem 6 directly yields Theorem 1 – the solution from Algorithm 1 converges to the optimal solution of Problem (5) asymptotically.

# 6 Empirical Evaluation

We report some numerical results in this section. Due to space constraints, more results, including those of subspace tracking, are deferred in the supplementary material.

## 6.1 Medium-scale Robust PCA

We here evaluate the ability of the proposed OR-PCA of correctly recovering the subspace of corrupted observations, under various settings of the intrinsic subspace dimension and error density. In particular, we adopt the batch robust PCA method, Principal Component Pursuit [4], as the batch

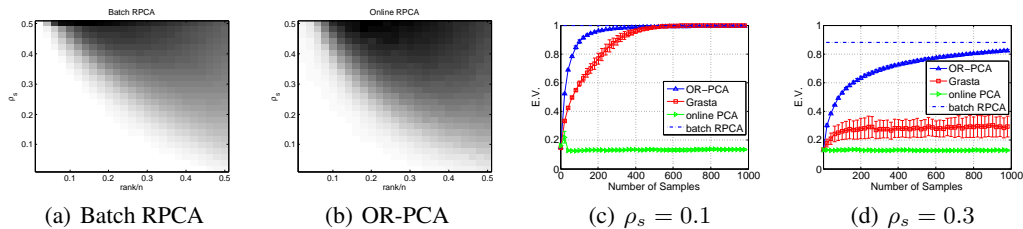

| (a) Batch RPCA | (b) OR-PCA | (c) $\rho_s = 0.1$ | (d) $\rho_s = 0.3$ |

Figure 1: (a) and (b): subspace recovery performance under different corruption fraction $\rho_s$ (vertical axis) and $\mathrm{rank}/n$ (horizontal axis). Brighter color means better performance; (c) and (d): the performance comparison of the OR-PCA, Grasta, and online PCA methods against the number of revealed samples under two different corruption levels $\rho_s$ with PCP as reference.

counterpart of the proposed OR-PCA method for reference. PCP estimates the subspace in a batch manner through solving the problem in (1) and outputs the low-rank data matrix. For fair comparison, we follow the data generation scheme of PCP as in [4]: we generate a set of $n$ clean data points as a product of $X = UV^T$, where the sizes of $U$ and $V$ are $p \times r$ and $n \times r$ respectively. The elements of both $U$ and $V$ are i.i.d. sampled from the $\mathcal{N}(0, 1/n)$ distribution. Here $U$ is the basis of the subspace and the intrinsic dimension of the subspace spanned by $U$ is $r$. The observations are generated through $Z = X + E$, where $E$ is a sparse matrix with a fraction of $\rho_s$ non-zero elements. The elements in $E$ are from a uniform distribution over the interval of $[-1000, 1000]$. Namely, the matrix $E$ contains gross but sparse errors.

We run the OR-PCA and the PCP algorithms 10 times under the following settings: the ambient dimension and number of samples are set as $p = 400$ and $n = 1,000$; the intrinsic rank $r$ of the subspace varies from 4 to 200; the value of error fraction, $\rho_s$, varies from very sparse 0.01 to relatively dense 0.5. The trade-off parameters of OR-PCA are fixed as $\lambda_1 = \lambda_2 = 1/\sqrt{p}$. The performance is evaluated by the similarity between the subspace obtained from the algorithms and the groundtruth. In particular, the similarity is measured by the Expressed Variance (E.V.) (see definition in [24]). A larger value of E.V. means better subspace recovery.

We plot the averaged E.V. values of PCP and OR-PCA under different settings in a matrix form, as shown in Figure 1(a) and Figure 1(b) respectively. The results demonstrate that under relatively low intrinsic dimension (small $\mathrm{rank}/n$) and sparse corruption (small $\rho_s$), OR-PCA is able to recover the subspace nearly perfectly (E.V.= 1). We also observe that the performance of OR-PCA is close to that of the PCP. This demonstrates that the proposed OR-PCA method achieves comparable performance with the batch method and verifies our convergence guarantee on the OR-PCA. In the relatively difficult setting (high intrinsic dimension and dense error, shown in the top-right of the matrix), OR-PCA performs slightly worse than the PCP, possibly because the number of streaming samples is not enough to achieve convergence.

To better demonstrate the robustness of OR-PCA to corruptions and illustrate how the performance of OR-PCA is improved when more samples are revealed, we plot the performance curve of OR-PCA against the number of samples in Figure 1(c), under the setting of $p = 400$, $n = 1,000$, $\rho_s = 0.1$, $r = 80$, and the results are averaged from 10 repetitions. We also apply GRASTA [10] to solve this RPCA problem in an online fashion as a baseline. The parameters of GRASTA are set as the values provided in the implementation package provided by the authors. We observe that when more samples are revealed, both OR-PCA and GRASTA steadily improve the subspace recovery. However, our proposed OR-PCA converges much faster than GRASTA, possibly because in each iteration OR-PCA obtains the optimal closed-form solution to the basis updating subproblem while GRASTA only takes one gradient descent step. Observe from the figure that after 200 samples are revealed, the performance of OR-PCA is already satisfactory (E.V.> 0.8). However, for GRASTA, it needs about 400 samples to achieve the same performance. To show the robustness of the proposed OR-PCA, we also plot the performance of the standard online (or incremental) PCA [1] for comparison. This work focuses on developing online *robust* PCA. The non-robustness of (online) PCA is independent of used optimization method. Thus, we only compare with the basic online PCA method [1], which is enough for comparing robustness. The comparison results are given in Figure 1(c). We observe that as expected, the online PCA cannot recover the subspace correctly (E.V.$\approx$ 0.1), since standard PCA is fragile to gross corruptions. We then increase the corruption

level to $\rho_s = 0.3$, and plot the performance curve of the above methods in Figure 1(d). From the plot, it can be observed that the performance of GRASTA decreases severely (E.V.$\approx 0.3$) while OR-PCA still achieves E.V. $\approx 0.8$. The performance of PCP is around $0.88$. This result clearly demonstrates the robustness advantage of OR-PCA over GRASTA. In fact, from other simulation results under different settings of intrinsic rank and corruption level (see supplementary material), we observe that the GRASTA breaks down at $25\%$ corruption (the value of E.V. is zero). However, OR-PCA achieves a performance of E.V.$\approx 0.5$, even in presence of $50\%$ outlier corruption.

## 6.2 Large-scale Robust PCA

We now investigate the computational efficiency of OR-PCA and the performance for large scale data. The samples are generated following the same model as explained in the above subsection. The results are provided in Table 1. All of the experiments are implemented in a PC with 2.83GHz Quad CPU and 8GB RAM. Note that batch RPCA cannot process these data due to out of memory.

Table 1: The comparison of OR-PCA and GRASTA under different settings of sample size ($n$) and ambient dimensions ($p$). Here $\rho_s = 0.3, r = 0.1p$. The corresponding computational time (in $\times 10^3$ seconds) is shown in the top row and the E.V. values are shown in the bottom row correspondingly. The results are based on the average of $5$ repetitions and the variance is shown in the parentheses.

| $p$ | $1 \times 10^3$ | | | $1 \times 10^4$ | |
|---|---|---|---|---|---|
| $n$ | $1 \times 10^6$ | $1 \times 10^8$ | $1 \times 10^{10}$ | $1 \times 10^6$ | $1 \times 10^8$ |
| OR-PCA | 0.013(0.0004) | 1.312(0.082) | 139.233(7.747) | 0.633(0.047) | 15.910(2.646) |
| | 0.99(0.01) | 0.99(0.00) | 0.99(0.00) | 0.82(0.09) | 0.82(0.01) |
| GRASTA | 0.023(0.0008) | 2.137(0.016) | 240.271(7.564) | 2.514(0.011) | 252.630(2.096) |
| | 0.54(0.08) | 0.55(0.02) | 0.57(0.03) | 0.45(0.02) | 0.46(0.03) |

From the above results, we observe that OR-PCA is much more efficient and performs better than GRASTA. In fact, the computational time of OR-PCA is linear in the sample size and nearly linear in the ambient dimension. When the ambient dimension is large ($p = 1 \times 10^4$), OR-PCA is more efficient than GRASTA with an order magnitude efficiency enhancement. We then compare OR-PCA with batch PCP. In each iteration, batch PCP needs to perform an SVD plus a thresholding operation, whose complexity is $O(np^2)$. In contrast, for OR-PCA, in each iteration, the computational cost is $O(pr^2)$, which is independent of the sample size and linear in the ambient dimension. To see this, note that in step 2) of Algorithm 1, the computation complexity is $O(r^2 + pr + r^3)$. Here $O(r^3)$ is for computing $L^T L$. The complexity of step 3) is $O(r^2 + pr)$. For step 4) (*i.e.*, Algorithm 2), the cost is $O(pr^2)$ (updating each column of $L$ requires $O(pr)$ and there are $r$ columns in total). Thus the total complexity is $O(r^2 + pr + r^3 + pr^2)$. Since $p \gg r$, the overall complexity is $O(pr^2)$.

The memory cost is significantly reduced too. The memory required for OR-PCA is $O(pr)$, which is independent of the sample size. This is much smaller than the memory cost of the batch PCP algorithm ($O(pn)$), where $n \gg p$ for large scale dataset. This is quite important for processing *big data*. The proposed OR-PCA algorithm can be easily parallelized to further enhance its efficiency.

## 7 Conclusions

In this work, we develop an online robust PCA (OR-PCA) method. Different from previous batch based methods, the OR-PCA need not "remember" all the past samples and achieves much higher storage efficiency. The main idea of OR-PCA is to reformulate the objective function of PCP (a widely applied batch RPCA algorithm) by decomposing the nuclear norm to an explicit product of two low-rank matrices, which can be solved by a stochastic optimization algorithm. We provide the convergence analysis of the OR-PCA method and show that OR-PCA converges to the solution of batch RPCA asymptotically. Comprehensive simulations demonstrate the effectiveness of OR-PCA.

**Acknowledgments**

J. Feng and S. Yan are supported by the Singapore National Research Foundation under its International Research Centre @Singapore Funding Initiative and administered by the IDM Programme Office. H. Xu is partially supported by the Ministry of Education of Singapore through AcRF Tier Two grant R-265-000-443-112 and NUS startup grant R-265-000-384-133.

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
