[Supplementary Material]

# Online Robust PCA via Stochastic Optimization
# Supplementary Material

**Jiashi Feng**
ECE Department
National University of Singapore
jiashi@nus.edu.sg

**Huan Xu**
ME Department
National University of Singapore
mpexuh@nus.edu.sg

**Shuicheng Yan**
ECE Department
National University of Singapore
eleyans@nus.edu.sg

## 1 Algorithm solving Problem (7)

For the data projection $\mathbf{r}$ and noise estimation $\mathbf{e}$, we can get the closed-form solutions for them respectively, as shown in Algorithm 1.1. In particular, the closed-form solution to a projection to $\ell_1$-ball in updating $\mathbf{e}$ involves a soft thresholding operator $\mathcal{S}_\lambda[\cdot]$ Hale et al. (2008), which is defined as:

$$\mathcal{S}_\lambda[x] \triangleq \begin{cases} x - \lambda, & \text{if } x > \lambda, \\ x + \lambda, & \text{if } x < -\lambda, \\ 0, & \text{otherwise.} \end{cases}$$

And it is conducted element-wisely on the involved vectors. The optimization iteration is terminated when the following convergence criterion is met:

$$\max(\|\mathbf{r}_{k+1} - \mathbf{r}_k\|/\|\mathbf{z}\|, \|\mathbf{e}_{k+1} - \mathbf{e}_k\|/\|\mathbf{z}\|) < \varepsilon.$$

Here $\varepsilon$ is set as $1 \times 10^{-6}$ throughout the simulations.

The details of the algorithm are summarized as follows,

---

**Algorithm 1.1** Data Projection

---

**Input:** $L = [\mathbf{l}_1, \ldots, \mathbf{l}_r] \in \mathbb{R}^{p \times r}$ (input basis), $\mathbf{z} \in \mathbb{R}^p$, parameters $\lambda_1$ and $\lambda_2$.
$\mathbf{e} \leftarrow \mathbf{0}$.
**while** not converged **do**
   Update the coefficient $\mathbf{r}$:
$$\mathbf{r} \leftarrow (L^T L + \lambda_1 I)^{-1} L^T (\mathbf{z} - \mathbf{e}).$$

   Update the sparse error $\mathbf{e}$:
$$\mathbf{e} \leftarrow \mathcal{S}_{\lambda_2}[\mathbf{z} - L\mathbf{r}].$$

**end while**
**Return** $L$.

---

## 2 Proof of Lemma 1

**Lemma 1** [optimality condition of Problem (4)] $\mathbf{r}^\star \in \mathbb{R}^r$ and $\mathbf{e}^\star \in \mathbb{R}^p$ is a solution of Problem (4) if and only if

$$C_\Lambda(\mathbf{z}_\Lambda - \mathbf{e}_\Lambda^\star) = \lambda_2 \text{sign}(\mathbf{e}_\Lambda^\star),$$
$$|C_{\Lambda^c}(\mathbf{z}_{\Lambda^c} - \mathbf{e}_{\Lambda^c}^\star)| \leq \lambda_2, \text{ otherwise,}$$
$$\mathbf{r}^\star = (L^T L + \lambda_1 I)^{-1} L^T (\mathbf{z} - \mathbf{e}^\star),$$

where $C = I - L(L^T L + \lambda_1 I)^{-1} L^T$ and $C_\Lambda$ denotes the column vectors of matrix $C$ indexed by the set $\Lambda = \{j | \mathbf{e}^\star[j] \neq 0\}$ and $\Lambda^c$ denotes the complementary set of $\Lambda$. Moreover, the optimal solution is unique.

*Proof.* Denote the subgradient of $\|\mathbf{e}\|_1$ as $\partial \|\mathbf{e}\|_1$ and it is known that

$$\partial \|\mathbf{e}\|_1 = \{\mathbf{u} | \mathbf{u}_i = \text{sign}(\mathbf{e}_i) \text{ if } \mathbf{e}_i \neq 0, \text{ and } |\mathbf{e}_i| \leq 1 \text{ otherwise}\}.$$

The point $(\mathbf{r}^\star, \mathbf{e}^\star)$ is a global minimum of (5) if and only if the vector zero is in its subgradient at $(\mathbf{r}^\star, \mathbf{e}^\star)$:

$$\exists \mathbf{u} \in \partial \|\mathbf{e}^\star\|_1 \text{ such that } \mathbf{e}^\star + L\mathbf{r}^\star - \mathbf{z} + \lambda_2 \mathbf{u} = 0, \tag{2.1}$$
$$-L^T \mathbf{z} + L^T L \mathbf{r}^\star + L^T \mathbf{e}^\star + \lambda_1 \mathbf{r}^\star = 0. \tag{2.2}$$

From (2.2), we have,

$$\mathbf{r}^\star = (L^T L + \lambda_1 I)^{-1} L^T (\mathbf{z} - \mathbf{e}^\star).$$

This proves the third inequality in the lemma. Substituting back into (2.1) yields

$$\left(I - L(L^T L + \lambda_1 I)^{-1} L^T\right)(\mathbf{z} - \mathbf{e}^\star) = \lambda_2 \mathbf{u}, \text{ where } \mathbf{u} \in \partial \|\mathbf{e}^\star\|_1.$$

Define the matrix

$$C \triangleq I - L(L^T L + \lambda_1 I)^{-1} L^T.$$

According to Woodbury matrix identity, we have

$$C = \left(I + \frac{1}{\lambda_1} L^T L\right)^{-1}.$$

Thus $C$ is invertible. We then have

$$C(\mathbf{z} - \mathbf{e}^\star) = \lambda_2 \mathbf{u}, \text{ where } \mathbf{u} \in \partial \|\mathbf{e}^\star\|_1.$$

Let $\Lambda = \{j | \mathbf{e}^\star[j] \neq 0\}$ be the index set of nonzero elements of the optimal solution $\mathbf{e}^\star$. Then we can show that

$$C_\Lambda(\mathbf{z}_\Lambda - \mathbf{e}_\Lambda^\star) = \lambda_2 \text{sign}(\mathbf{e}_\Lambda^\star)$$

Here $C_\Lambda$ denotes submatrix of $C$ consisting of the column vectors of matrix $C$ indexed by $\Lambda$. Then we can solve out that

$$\mathbf{e}_\Lambda^\star = (C_\Lambda^T C_\Lambda)^{-1}(C_\Lambda \mathbf{z}_\Lambda - \lambda \text{sign}(\mathbf{e}_\Lambda^\star)),$$
$$\mathbf{e}_{\Lambda^c}^\star = 0.$$

Since $C$ is invertible, $C$ is column full rank. Thus $C_\Lambda$ is column full rank and $C_\Lambda^T C_\Lambda$ is invertible, the solution $\mathbf{e}^\star$ is unique and thus $\mathbf{r}^\star$ is also unique. $\square$

## 3 Proof of Lemma 2

**Lemma 2** Assume the observations $\mathbf{z}$ are always bounded. Define

$$\{\mathbf{r}^\star, \mathbf{e}^\star\} = \arg\min_{\mathbf{r}, \mathbf{e}} \frac{1}{2} \|\mathbf{z} - L\mathbf{r} - \mathbf{e}\|_2^2 + \frac{\lambda_1}{2} \|\mathbf{r}\|_2^2 + \lambda_2 \|\mathbf{e}\|_1.$$

Then, 1) the function $\ell$ defined in (4) is continuously differentiable and

$$\nabla_L \ell(\mathbf{z}, L) = (L\mathbf{r}^\star + \mathbf{e}^\star - \mathbf{z})\mathbf{r}^{\star T};$$

2) $\nabla f(L) = \mathbb{E}_\mathbf{z}[\nabla_L \ell(\mathbf{z}, L)]$; and 3)$\nabla_L f(L)$ is Lipschitz.

*Proof.* To reveal the regularity of the expected loss function $f$ and its derivative $\nabla f$, we need first to prove the regularity of the loss function $\ell$ as stated in the first claim.

**Proof of the first claim**

Define a function $\tilde{f}$ as

$$\tilde{f}(\mathbf{r}, \mathbf{e}, \mathbf{z}, L) \triangleq \frac{1}{2}\|\mathbf{z} - L\mathbf{r} - \mathbf{e}\|_2^2 + \frac{\lambda_1}{2}\|L\|_F^2 + \frac{\lambda_1}{2}\|\mathbf{r}\|_2^2 + \lambda_2\|\mathbf{e}\|_1.$$

Thus the loss function $\ell$ can be expressed as

$$\ell(\mathbf{z}, L) = \min_{\mathbf{r},\mathbf{e}} \tilde{f}(\mathbf{r}, \mathbf{e}, \mathbf{z}, L).$$

The function $\tilde{f}(\mathbf{r}, \mathbf{e}, \mathbf{z}, L)$ is continuous, and for all $\mathbf{r} \in \mathbb{R}^r, \mathbf{e} \in \mathbb{R}^p$, the function $\tilde{f}(\mathbf{r}, \mathbf{e}, \cdot, \cdot)$ is differentiable, and the derivative $\nabla_L \tilde{f}(\mathbf{r}, \mathbf{e}, \cdot, \cdot) = (L\mathbf{r} + \mathbf{e} - \mathbf{z})\mathbf{r}^T$ is continuous. Furthermore, according to Lemma 1, $\tilde{f}(\cdot, \cdot, \mathbf{z}, L)$ has unique minimizer $(\mathbf{r}^\star, \mathbf{e}^\star)$, thus Lemma 3 directly applies and we obtain that $\ell(\mathbf{z}, L)$ is differentiable in $L$ and

$$\nabla_L \ell(\mathbf{z}, L) = \nabla_L \tilde{f}(\mathbf{r}^\star, \mathbf{e}^\star, \mathbf{z}, L) = (L\mathbf{r}^\star + \mathbf{e}^\star - \mathbf{z})\mathbf{r}^{\star T} + \lambda_1 L.$$

Thus, we complete the proof of the first claim.

**Proof of the second claim**

According to the first claim, the function $\ell_{\mathbf{z},L}$ is continuously differentiable, thus

$$\nabla_L f(L) = \nabla_L \mathbb{E}_{\mathbf{z}}[\ell(\mathbf{z}, L)] = \mathbb{E}_{\mathbf{z}}[\nabla_L \ell(\mathbf{z}, L)].$$

Equipped with the above two results, we are ready to prove that the derivative $\nabla_L f(L)$ is Lipschitz.

**Proof of the third claim**

To prove that $\nabla f(L)$ is Lipschitz, we will show that for all bounded observations $\mathbf{z}$, $\mathbf{r}^\star(\mathbf{z}, \cdot)$ and $\mathbf{e}^\star(\mathbf{z}, \cdot)$ are Lipschitz with constants independent of $\mathbf{z}$. First, the loss function $\ell(\mathbf{z}, L)$ defined in (4) is continuous in $\mathbf{r}, \mathbf{e}, L, \mathbf{z}$ and has a unique minimum (according to Lemma 1) for fixed $\mathbf{z}$ and $L$, thus the optimal solutions $\mathbf{r}^\star$ and $\mathbf{e}^\star$ are continuous in $L$ and $\mathbf{z}$.

Consider a matrix $L$ and a sample $\mathbf{z}$, and denote $\mathbf{r}^\star$ and $\mathbf{e}^\star$ as the corresponding optimal solutions. Denote by $\Lambda$ the set of the indices such that $|C_\Lambda(\mathbf{z}_\Lambda - \mathbf{e}_\Lambda^\star)| = \lambda_1$ (see Lemma 1). Here the matrix $C$ is defined as $C = I - L(L^T L + \lambda_1 I)^{-1}L^T$. Since $C_\Lambda$ is nonsingular, $C_\Lambda(\mathbf{z}_\Lambda - \mathbf{e}_\Lambda^\star)$ is continuous in $L$ and $\mathbf{z}$. Thus we consider a small perturbation of $(\mathbf{z}, L)$ in one of their open neighborhood $V$, such that for all $(\mathbf{z}', L')$ in $V$, we have if $j \notin \Lambda$, $\left|C_j'(\mathbf{z}'[j] - \mathbf{e}^{\star\prime}[j])\right| < \lambda_2$ and $\mathbf{e}^{\star\prime}[j] = 0$, where $\mathbf{e}^{\star\prime} = \mathbf{e}^\star(\mathbf{z}', L')$. Namely the support set of $\mathbf{e}^\star$ is not changed.

Based on the about continuity, we consider the following function

$$\tilde{\ell}(\mathbf{z}_\Lambda, L_\Lambda, \mathbf{r}, \mathbf{e}_\Lambda) \triangleq \frac{1}{2}\|\mathbf{z}_\Lambda - L_\Lambda\mathbf{r} - \mathbf{e}_\Lambda\|_2^2 + \frac{\lambda_1}{2}\|L_\Lambda\|_F^2 + \frac{\lambda_1}{2}\|\mathbf{r}\|_2^2 + \lambda_2\|\mathbf{e}_\Lambda\|_1.$$

Since the Hessian matrix of the function $\tilde{\ell}(\mathbf{z}_\Lambda, L_\Lambda, \cdot, \cdot)$ w.r.t. $\mathbf{r}$, $I \otimes (L_\Lambda^T L_\Lambda + \lambda_1 I)$, and the Hessian matrix w.r.t. $\mathbf{e}_\Lambda$, $I \otimes \lambda_2 I$, are positive definite, we have the function $\tilde{\ell}(\mathbf{z}_\Lambda, L_\Lambda, \cdot, \cdot)$ is strictly convex and

$$
\begin{aligned}
&\tilde{\ell}(\mathbf{z}_\Lambda, L_\Lambda, \mathbf{r}^{\star\prime}, \mathbf{e}_\Lambda^{\star\prime}) - \tilde{\ell}(\mathbf{z}_\Lambda, L_\Lambda, \mathbf{r}^\star, \mathbf{e}_\Lambda^\star) \\
\geq\ & \lambda_1\|\mathbf{r}^{\star\prime} - \mathbf{r}^\star\|_2^2 + \lambda_2\|\mathbf{e}_\Lambda^{\star\prime} - \mathbf{e}_\Lambda^\star\|_2^2 \\
\geq\ & \min(\lambda_1, \lambda_2)(\|\mathbf{r}^{\star\prime} - \mathbf{r}^\star\|_2^2 + \|\mathbf{e}_\Lambda^{\star\prime} - \mathbf{e}_\Lambda^\star\|_2^2). \hspace{3em} (3.1)
\end{aligned}
$$

We then show that the function $\tilde{\ell}(\mathbf{z}, L, \cdot, \cdot) - \tilde{\ell}(\mathbf{z}', L', \cdot, \cdot)$ is Lipschitz continuous.

To this end, we calculate the difference of the above function:

$$
\begin{aligned}
&\left(\tilde{\ell}(\mathbf{z}, L, \mathbf{r}, \mathbf{e}) - \tilde{\ell}(\mathbf{z}', L', \mathbf{r}, \mathbf{e})\right) - \left(\tilde{\ell}(\mathbf{z}, L, \mathbf{r}', \mathbf{e}') - \tilde{\ell}(\mathbf{z}', L', \mathbf{r}', \mathbf{e}')\right) \\
=\ & \frac{1}{2}\left(\|\mathbf{z} - L\mathbf{r} - \mathbf{e}\|_2^2 - \|\mathbf{z}' - L'\mathbf{r} - \mathbf{e}\|_2^2\right) - \frac{1}{2}\left(\|\mathbf{z} - L\mathbf{r}' - \mathbf{e}'\|_2^2 - \|\mathbf{z}' - L'\mathbf{r}' - \mathbf{e}'\|_2^2\right)
\end{aligned}
$$

Define a matrix $A = [L, I]$ and a vector $\mathbf{b} = [\mathbf{r}; \mathbf{e}]$, and we have $L\mathbf{r} + \mathbf{e} = A\mathbf{b}$. Then,

$$\left(\tilde{\ell}(\mathbf{z}, L, \mathbf{r}, \mathbf{e}) - \tilde{\ell}(\mathbf{z}', L', \mathbf{r}, \mathbf{e})\right) - \left(\tilde{\ell}(\mathbf{z}, L, \mathbf{r}', \mathbf{e}') - \tilde{\ell}(\mathbf{z}', L', \mathbf{r}', \mathbf{e}')\right)$$

$$= \frac{1}{2}\left(\|\mathbf{z} - A\mathbf{b}\|_2^2 - \|\mathbf{z}' - A'\mathbf{b}\|_2^2\right) - \frac{1}{2}\left(\|\mathbf{z} - A\mathbf{b}'\|_2^2 - \|\mathbf{z}' - A'\mathbf{b}'\|_2^2\right)$$

It is easy to show that the function $\|\mathbf{z} - A\mathbf{b}\|_2^2 - \|\mathbf{z}' - A'\mathbf{b}\|_2^2$ is Lipschitz with constant as $c_1\|A - A'\|_F + c_2\|\mathbf{z} - \mathbf{z}'\|_2$, where $c_1, c_2$ are constants independent of $A, A', \mathbf{z}, \mathbf{z}'$. Thus,

$$\left(\tilde{\ell}(\mathbf{z}, L, \mathbf{r}, \mathbf{e}) - \tilde{\ell}(\mathbf{z}', L', \mathbf{r}, \mathbf{e})\right) - \left(\tilde{\ell}(\mathbf{z}, L, \mathbf{r}', \mathbf{e}') - \tilde{\ell}(\mathbf{z}', L', \mathbf{r}', \mathbf{e}')\right)$$

$$\leq \left(c_1\|A - A'\|_F + c_2\|\mathbf{z} - \mathbf{z}'\|_2\right)\|\mathbf{b} - \mathbf{b}'\|_2$$

$$= \left(c_1\|L - L'\|_F + c_2\|\mathbf{z} - \mathbf{z}'\|_2\right)\left(\|\mathbf{r} - \mathbf{r}'\|_2 + \|\mathbf{e} - \mathbf{e}'\|_2\right)$$

According to (3.1) in the supplementary material, and considering $(\mathbf{r}^{*\prime}, \mathbf{e}_\Lambda^{*\prime})$ minimizes the loss $\tilde{\ell}(\mathbf{z}', L', \cdot, \cdot)$, we have

$$\min(\lambda_1, \lambda_2)\left(\|\mathbf{r}^{*\prime} - \mathbf{r}^*\|_2^2 + \|\mathbf{e}_\Lambda^{*\prime} - \mathbf{e}_\Lambda^*\|_2^2\right)$$

$$\leq \tilde{\ell}(\mathbf{z}_\Lambda, L_\Lambda, \mathbf{r}^{*\prime}, \mathbf{e}_\Lambda^{*\prime}) - \tilde{\ell}(\mathbf{z}_\Lambda, L_\Lambda, \mathbf{r}^*, \mathbf{e}_\Lambda^*)$$

$$= \tilde{\ell}(\mathbf{z}_\Lambda, L_\Lambda, \mathbf{r}^{*\prime}, \mathbf{e}_\Lambda^{*\prime}) - \tilde{\ell}(\mathbf{z}_\Lambda', L_\Lambda', \mathbf{r}^*, \mathbf{e}_\Lambda^*) + \tilde{\ell}(\mathbf{z}_\Lambda', L_\Lambda', \mathbf{r}^*, \mathbf{e}_\Lambda^*) - \tilde{\ell}(\mathbf{z}_\Lambda, L_\Lambda, \mathbf{r}^*, \mathbf{e}_\Lambda^*)$$

$$\leq \tilde{\ell}(\mathbf{z}_\Lambda, L_\Lambda, \mathbf{r}^{*\prime}, \mathbf{e}_\Lambda^{*\prime}) - \tilde{\ell}(\mathbf{z}_\Lambda', L_\Lambda', \mathbf{r}^{*\prime}, \mathbf{e}_\Lambda^{*\prime}) + \tilde{\ell}(\mathbf{z}_\Lambda', L_\Lambda', \mathbf{r}^*, \mathbf{e}_\Lambda^*) - \tilde{\ell}(\mathbf{z}_\Lambda, L_\Lambda, \mathbf{r}^*, \mathbf{e}_\Lambda^*)$$

$$\leq \left(c_1\|L_\Lambda - L_\Lambda'\|_F + c_2\|\mathbf{z}_\Lambda - \mathbf{z}_\Lambda'\|_2\right)\left(\|\mathbf{r}^{*\prime} - \mathbf{r}^*\|_2 + \|\mathbf{e}_\Lambda^{*\prime} - \mathbf{e}_\Lambda^*\|_2\right).$$

Therefore, we have,

$$\left(\|\mathbf{r}^{*\prime} - \mathbf{r}^*\|_2 + \|\mathbf{e}_\Lambda^{*\prime} - \mathbf{e}_\Lambda^*\|_2\right) \leq \frac{1}{\min(\lambda_1, \lambda_2)}\left(c_1\|L_\Lambda - L_\Lambda'\|_F + c_2\|\mathbf{z}_\Lambda - \mathbf{z}_\Lambda'\|_2\right).$$

Combining the second claim, we can conclude the third claim.

$\square$

## 4 Proof of Theorem 2

**Theorem 2** [Convergence of the surrogate function $g_t$] Let $g_t$ denote the surrogate function defined in (6). Then, $g_t(L_t)$ converges almost surely when the solution $L_t$ is given by Algorithm 1.

*Proof.* We prove the convergence of the sequence $g_t(L_t)$ by showing that the stochastic positive process

$$u_t \triangleq g_t(L_t) \geq 0,$$

is a quasi-martingale Fisk (1965). According to Lemma 4, if the sum of the positive difference of $u_t$ is bounded, $u_t$ is a quasi-martingale. And the sum converges almost surely. Thus, we compute the difference of $u_t$ and obtain

$$u_{t+1} - u_t$$

$$= g_{t+1}(L_{t+1}) - g_t(L_t)$$

$$= g_{t+1}(L_{t+1}) - g_{t+1}(L_t) + g_{t+1}(L_t) - g_t(L_t)$$

$$= g_{t+1}(L_{t+1}) - g_{t+1}(L_t) + \frac{\ell(\mathbf{z}_{t+1}, L_t) - f_t(L_t)}{t + 1} + \frac{f_t(L_t) - g_t(L_t)}{t + 1}. \qquad (4.1)$$

Here the third equality is from the fact that $g_{t+1}(L_t) = \frac{1}{t+1}\ell(\mathbf{z}_{t+1}, L_t) + \frac{t}{t+1}g_t(L_t)$. Since $L_{t+1}$ minimizes $g_{t+1}$, $g_{t+1}(L_{t+1}) - g_{t+1}(L_t) \leq 0$. Since the surrogate $g_t$ upperbounds the empirical cost $f_t$, $g_t \geq f_t$, we also have $f_t(L_t) - g_t(L_t) \leq 0$. Thus we have

$$u_{t+1} - u_t \leq \frac{\ell(\mathbf{z}_{t+1}, L_t) - f_t(L_t)}{t + 1}.$$

We then have

$$\mathbb{E}[u_{t+1} - u_t | \mathcal{F}_t] \leq \frac{\mathbb{E}[\ell(\mathbf{z}_{t+1}, L_t)|\mathcal{F}_t] - f_t(L_t)}{t+1} = \frac{f(L_t) - f_t(L_t)}{t+1} \leq \frac{\|f - f_t\|_\infty}{t+1}. \qquad (4.2)$$

Here $\|f - f_t\|_\infty = \sup_{f \in F} |f - f_t|$ and $F = \{\ell(\mathbf{z}, L) : \mathcal{Z} \to \mathbb{R}, L \in \mathcal{L}\}$. To bound $\mathbb{E}[\sqrt{t}\|f - f_t\|_\infty]$, here we use the Lemma 5. It is easy to show that in our case, all the hypotheses are verified, namely, $\ell(\mathbf{z}, \cdot)$ is uniformly Lipschitz and bounded (see Lemma 2). Thus $\mathbb{E}_{\mathbf{z}}[\ell(\mathbf{z}, L)^2]$ exists and is uniformly bounded. Therefore, Lemma 5 applies and there exists a constant $\kappa > 0$ such that

$$\mathbb{E}[\sqrt{t}|f - f_t|_\infty] \leq \kappa.$$

Therefore,

$$\mathbb{E}[\mathbb{E}[u_{t+1} - u_t | \mathcal{F}_t]^+] = \mathbb{E}\{\max(\mathbb{E}[u_{t+1} - u_t | \mathcal{F}], 0)\} \leq \frac{\kappa}{t^{\frac{3}{2}}}.$$

Therefore,

$$\mathbb{E}[\mathbb{E}[u_{t+1} - u_t | \mathcal{F}_t]^+] \leq \frac{\kappa}{t^{\frac{3}{2}}}.$$

Therefore, defining $\delta_t$ as in Lemma 4:

$$\delta_t = \begin{cases} 1, & \text{if } \mathbb{E}[u_{t+1} - u_t | \mathcal{F}_t] > 0, \\ 0, & \text{otherwise,} \end{cases}$$

we have

$$\sum_{t=1}^{\infty} \mathbb{E}[\delta_t(u_{t+1} - u_t)] = \sum_{t=1}^{\infty} \mathbb{E}[\mathbb{E}[u_{t+1} - u_t | \mathcal{F}_t]^+] \leq \sum_{t=1}^{\infty} \frac{\kappa}{t^{\frac{3}{2}}} \leq +\infty.$$

Thus, we can apply Lemma 4, which proves that $u_t = g_t$ converges almost surely and that

$$\sum_{t=1}^{\infty} |\mathbb{E}[u_{t+1} - u_t | \mathcal{F}_t]| < +\infty \quad a.s.$$

Thus we complete the proof. □

## 5 Proof of Theorem 3

**Theorem 3** [Difference of the solution $L_t$] For the two successive solutions obtained from Algorithm 1, we have

$$\|L_{t+1} - L_t\|_F = O(1/t) \quad a.s.$$

*Proof.* The Hessian matrix of $g_t(L)$ is $H = I \otimes (A_t + \lambda_1 I)$. Here $\otimes$ denotes the Kronecker production and $A_t = \sum_{i=1}^{t} \mathbf{r}_i \mathbf{r}_i^T$. The smallest eigenvalue of $H$ is equal to the smallest eigenvalue of matrix $(A_t + \lambda_1 I)$, which must be larger than $\lambda_1$ since $A_t$ is a semi-definite positive matrix. Thus $g_t(L)$ is strictly convex. And we have,

$$g_t(L_{t+1}) - g_t(L_t) \geq \lambda_1 \|L_{t+1} - L_t\|_F^2. \qquad (5.1)$$

Since $g_{t+1}(L_{t+1}) < g_{t+1}(L_t)$ due to $L_{t+1}$ minimizing $g_{t+1}$, we have

$$g_t(L_{t+1}) - g_t(L_t) \leq g_t(L_{t+1}) - g_{t+1}(L_{t+1}) + g_{t+1}(L_t) - g_t(L_t) = v_t(L_{t+1}) - v_t(L_t).$$

Here we define $v_t(L) \triangleq g_t(L) - g_{t+1}(L)$. And we have,

$$\nabla_L v_t(L) = \nabla_L g_t(L_t) - \nabla_L g_{t+1}(L_t) = \frac{1}{t}(L\tilde{A}_t - B_t) - \frac{1}{t+1}(L\tilde{A}_{t+1} - B_{t+1}).$$

Here $\tilde{A} \triangleq A + \lambda_1 I$ as defined in Algorithm 2. Therefore, by utilizing the triangle inequality and $\|AB\|_F \leq \|A\|_F \|B\|_F$, we can obtain,

$$\begin{aligned}
\|\nabla_L v_t(L)\|_F &= \left\| \frac{1}{t} L \left( \tilde{A}_t - \frac{t}{t+1} \tilde{A}_{t+1} \right) - \frac{1}{t} \left( B_t - \frac{t}{t+1} B_{t+1} \right) \right\|_F \\
&\leq \frac{1}{t} \left( \|L\|_F \left\| \tilde{A}_t - \frac{t\tilde{A}_{t+1}}{t+1} \right\|_F + \left\| B_t - \frac{tB_{t+1}}{t+1} \right\|_F \right).
\end{aligned}$$

Since the basis $L$ is usually bounded $\|L\|_F < \kappa_1$ (Assumption 1), the function $v_t(L)$ is Lipschitz with constant $c_t = \frac{1}{t}\left(\kappa_1\|\tilde{A}_t - \frac{t\tilde{A}_{t+1}}{t+1}\|_F + \|B_t - \frac{tB_{t+1}}{t+1}\|_F\right)$. Thus, we have

$$g_t(L_{t+1}) - g_t(L_t) \le v_t(L_{t+1}) - v_t(L_t) \le c_t\|L_{t+1} - L_t\|_F.$$

Substituting into (5.1), we can then obtain that

$$\|L_{t+1} - L_t\|_F \le \frac{c_t}{\lambda_1}.$$

Since $c_t = O(1/t)$, we have $\|L_{t+1} - L_t\|_F = O(1/t)$. $\qquad\qquad\qquad\qquad\qquad\qquad\qquad\square$

## 6 Proof of Theorem 4

**Theorem 4** [Convergence of $f$] Let $g_t$ denote the surrogate function defined in (2). Then, 1) $f(L_t) - g_t(L_t)$ converges almost surely to 0; and 2) $f(L_t)$ converges almost surely, when the solution $L_t$ is given by Algorithm 1.

*Proof.* From (4.1), we can obtain that

$$\frac{g_t(L_t) - f_t(L_t)}{t+1} \le \frac{\ell(\mathbf{z}_{t+1}, L_t) - f_t(L_t)}{t+1} - (g_{t+1} - g_t) \le \frac{\ell(\mathbf{z}_{t+1}, L_t) - f_t(L_t)}{t+1} + [g_{t+1} - g_t]^-$$

Taking the conditional expectation on the filtration $\mathcal{F}_t$ as in the proof of Theorem 2, we obtain

$$\mathbb{E}\left[\frac{g_t(L_t) - f_t(L_t)}{t+1}\Big|\mathcal{F}_t\right] = \frac{g_t(L_t) - f_t(L_t)}{t+1} \le \mathbb{E}\left[\frac{\ell(\mathbf{z}_{t+1}, L_t) - f_t(L_t)}{t+1}\Big|\mathcal{F}_t\right] + \mathbb{E}[[g_{t+1} - g_t]^-|\mathcal{F}_t].$$

$$\sum_{t=1}^{\infty}\frac{g_t(L_t) - f_t(L_t)}{t+1}$$

$$\le \sum_{t=1}^{\infty}\mathbb{E}\left[\frac{\ell(\mathbf{z}_{t+1}, L_t) - f_t(L_t)}{t+1}\Big|\mathcal{F}_t\right] + \sum_{t=1}^{\infty}\mathbb{E}[[g_{t+1} - g_t]^-|\mathcal{F}_t]$$

$$\le \sum_{t=1}^{\infty}\frac{|f - f_t|}{t+1} + \sum_{t=1}^{\infty}\mathbb{E}[[g_{t+1} - g_t]^-|\mathcal{F}_t].$$

Here $[\cdot]^-$ means taking negative part. The second inequality is from (4.2). According to Theorem 2, the function $g_t$ converges almost surely. And we have

$$\sum_{t=1}^{\infty}|\mathbb{E}[[g_{t+1} - g_t]^+]|\mathcal{F}_t| < +\infty \;\; a.s.$$

By symmetry we can also obtain similarly

$$\sum_{t=1}^{\infty}|\mathbb{E}[[g_{t+1} - g_t]^-]|\mathcal{F}_t| < +\infty \;\; a.s.$$

According to central limit theorem, we have $\sqrt{t}|f - f_t|$ converges almost surely when $t \to \infty$. Therefore $\sum_{t=1}^{\infty}\frac{|f-f_t|}{t+1}$ converges almost surely. Then we obtain the almost sure convergence of the positive sum

$$\sum_{t=1}^{\infty}\frac{g_t(L_t) - f_t(L_t)}{t+1} \le \sum_{t=1}^{\infty}\frac{|f - f_t|}{t+1} + \sum_{t=1}^{\infty}|\mathbb{E}[u_{t+1} - u_t|\mathcal{F}_t]| \le \infty.$$

Since both $g_t$ and $f_t$ are Lipschitz continuous, there exists a constant $\kappa' > 0$ such that

$$|g_{t+1}(L_{t+1}) - f_{t+1}(L_{t+1}) - (g_t(L_t) - f_t(L_t))| \le \kappa'\|L_{t+1} - L_t\|_F.$$

According to Theorem 3, $\|L_{t+1} - L_t\|_F = O(1/t)$. Thus it is easy to verify that the hypotheses of Lemma 6 are satisfied. Therefore,

$$g_t(L_t) - f_t(L_t) \xrightarrow[t \to +\infty]{} 0 \ \ a.s.$$

Since $g_t(L_t)$ converges almost surely, this shows that $f_t(L_t)$ converges almost surely to the same limit. Note that we have in addition $\|f_t - f\|_\infty \xrightarrow[t \to +\infty]{} 0$ a.s. Therefore,

$$g_t(L_t) - f(L_t) \xrightarrow[t \to +\infty]{} 0 \ \ a.s.$$

and $f(L_t)$ converges almost surely. $\qquad\square$

# 7 Proof of Theorem 5

**Theorem 5** The first order optimal condition for the minimization of the objective function in (6) is satisfied by $L_t$, the solution provided by Algorithm 1, when $t$ tends to infinity.

*Proof.* Since the function $g_t$ converges almost surely (see Theorem 2), $g_t = \text{Tr}(L^T L \tilde{A}_t/t) - \text{Tr}(L^T B_t/t)$, thus the sequences of matrices $\tilde{A}_t/t, B_t/t$ are bounded. It is possible to extract converging subsequences. Let us assume for a moment that these sequences converge respectively to two matrices $A_\infty$ and $B_\infty$. In that case, $L_t$ converges to a matrix $L_\infty$. Let $U$ be a matrix in $\mathbb{R}^{p \times r}$. Since $g_t$ upperbounds $f_t$ on $\mathbb{R}^{p \times r}$, for all $t$,

$$g_t(L_t + U) \geq f_t(L_t + U).$$

Taking the limit when $t$ tends to infinity,

$$g_\infty(L_\infty + U) \geq f(L_\infty + U).$$

Let $h_t > 0$ be a sequence that converges to 0. Using a first order Taylor expansion, and using the fact that $\nabla f$ is Lipschitz (see Lemma 3) and $g_\infty(L_\infty) = f(L_\infty)$ a.s. (see Theorem 4), we have

$$f(L_\infty) + \text{Tr}(h_t L^T \nabla g_\infty(L_\infty)) + o(h_t L) \geq f(L_\infty) + \text{Tr}(h_t L^T \nabla f(L_\infty)) + o(h_t L),$$

and it follows that

$$\text{Tr}\left(\frac{1}{\|L\|_F} L^T \nabla g_\infty(L_\infty)\right) \geq \text{Tr}\left(\frac{1}{\|L\|_F} L^T \nabla f(L_\infty)\right).$$

Since the above inequality is true for all $L$, we have $\nabla g_\infty(L_\infty) = \nabla f(L_\infty)$. Since the first-order necessary condition for $L_\infty$ being an optimum of $g_\infty$ is that $\nabla g_\infty = 0$. Thus at $L_\infty$, we have $\nabla f(L_\infty) = 0$. Namely, the first-order optimum condition for $f$ at $L_\infty$ is also verified. $\qquad\square$

# 8 Proof of Theorem 6

**Theorem 6** When the solution $L$ satisfies the first order condition for minimizing the objective function in (5), the obtained solution $L$ is the optimal solution of the problem (5) if $L$ is full rank.

*Proof.* The minimization of the objective function in (6),

$$\min_L \lim_{n \to \infty} \frac{1}{n} \sum_{i=1}^n \ell(\mathbf{z}_i, L)$$

is equivalent to

$$\min_{L,R,E} \frac{1}{2} \|Z - LR^T - E\|_F^2 + \frac{\lambda_1}{2} \left(\|L\|_F^2 + \|R\|_F^2\right) + \lambda_2 \|E\|. \tag{8.1}$$

Here $Z = [\mathbf{z}_1, \dots, \mathbf{z}_n], R = [\mathbf{r}_1^T; \dots; \mathbf{r}_n^T]$ and $E = [\mathbf{e}_1, \dots, \mathbf{e}_n]$.

When the first order optimal condition is satisfied, we have

$$(LR^T - \tilde{Z})R + \lambda_1 L = 0, \tag{8.2}$$

$$(RL^T - \tilde{Z}^T)L + \lambda_1 R = 0, \tag{8.3}$$

$$LR^T - \tilde{Z} \in \lambda_2 \partial \|E\|_1. \tag{8.4}$$

Here $\tilde{Z} \triangleq Z - E$. Note that for any invertible matrix $Q$, the pair $(LQ, RQ^{-1})$ provides a factorization equivalent to $(L, R)$. In particular, any solution $(L, R)$ can be orthogonalized to a (non-unique) equivalent orthogonal solution $\bar{L} = LQ, \bar{R} = RQ^{-1}$ such that $\bar{R}^T \bar{R} = \Lambda_R$ and $\bar{L}^T \bar{L} = \Lambda_L$ are diagonal matrices Srebro & Jaakkola (2003). Substituting $\bar{R}^T \bar{R} = \Lambda_R$ and $\bar{L}^T \bar{L} = \Lambda_L$ into (8.2) and (8.3), we can obtain that $\Lambda_L = \Lambda_R = \Lambda$.

Since we can always perform the orthgonalization operation on the obtained solution $L$ and $R$, we focus on an orthogonal solution, where $R^T R = \Lambda \in \mathbb{R}^{r \times r}$ and $L^T L = \Lambda \in \mathbb{R}^{r \times r}$ . Since $L$ and $R$ are full rank, the elements in the diagonal of matrix $\Lambda$ are non-zero.

From (8.2) we can obtain

$$L = \tilde{Z}R(R^T R + \lambda_1 I)^{-1} = \tilde{Z}R(\Lambda + \lambda_1 I)^{-1}. \tag{8.5}$$

Substituting back into (8.3), we have

$$R\Lambda - \tilde{Z}^T L + \lambda_1 R = 0.$$

Namely,

$$R\Lambda - \tilde{Z}^T \tilde{Z}R(\Lambda + \lambda_1 I)^{-1} + \lambda_1 R = 0,$$

$$R(\Lambda + \lambda_1 I)^2 = \tilde{Z}^T \tilde{Z}R. \tag{8.6}$$

Define $R' \triangleq R(\sqrt{\Lambda})^{-1}$, then we have $R'^T R' = (\sqrt{\Lambda})^{-1} R^T R (\sqrt{\Lambda})^{-1} = I$. Namely, the matrix $R'$ is an orthogonal matrix. From the above equation, we conclude that

$$R'\sqrt{\Lambda}(\Lambda + \lambda_1 I)^2 = \tilde{Z}^T \tilde{Z}R'\sqrt{\Lambda}.$$

$$R'(\Lambda + \lambda_1 I)^2 = \tilde{Z}^T \tilde{Z}R'.$$

Therefore, the columns of the matrix $R'$ are the eigenvectors of the matrix $\tilde{Z}^T \tilde{Z}$ . Thus the columns of the matrix $R$ are the eigenvectors of the matrix $\tilde{Z}^T \tilde{Z}$ scaled by the square root of the matrix $\Lambda$. And the eigenvalues of the matrix $\tilde{Z}^T \tilde{Z}$ are the elements in the diagonal of matrix $(\Lambda + \lambda_1 I)^2$.

From (8.5) we have

$$\tilde{Z}\tilde{Z}^T L = \tilde{Z}\tilde{Z}^T \tilde{Z}R(\Lambda + \lambda_1 I)^{-1} \stackrel{(8.6)}{=} \tilde{Z}R(\Lambda + \lambda_1 I) \stackrel{(8.5)}{=} L(\Lambda + \lambda_1 I)^2.$$

Thus similar to $R$, the columns of matrix $L$ correspond to the eigenvectors of the matrix $\tilde{Z}\tilde{Z}^T$ scaled by the square root of the matrix $\Lambda$.

Performing SVD on the matrix $\tilde{Z}$ provides $\tilde{Z} = U\Sigma V^T = U_1 \Sigma_1 V_1^T + U_2 \Sigma_2 V_2^T$. Here $U_1^T U_2 = 0$, $V_1^T V_2 = 0$ and $\Sigma_1 \in \mathbb{R}^{k \times k}, \Sigma_2 \in \mathbb{R}^{(n-k) \times (n-k)}$.

From the above results, we can obtain $L = U_1 \sqrt{\Lambda}$ and $R = V_1 \sqrt{\Lambda}$.

$$\tilde{Z}^T \tilde{Z} = V\Sigma^2 V^T.$$

Thus

$$\Sigma_1 = \Lambda + \lambda_1 I.$$

Since the matrix $L$ is full rank, $L^T L = \Lambda$ is positive definite. Thus $\Sigma_1 \succ \lambda_1 I$.

The obtained solution $X = LR^T = U_1 \Lambda V_1^T = U_1(\Sigma_1 - \lambda_1 I)V_1^T$. We can obtain that

$$\tilde{Z} - X = U\Sigma V^T - U_1(\Sigma_1 - \lambda_1 I)V_1^T = \lambda_1 U_1 V_1^T + U_2 \Sigma_2 V_2^T = \lambda_1(U_1 V_1^T + W),$$

where $W = U_2 \Sigma_2 V_2^T / \lambda_1$.

Thus, it is easy to verify that

$$\tilde{Z} - X = Z - E - X \in \partial \lambda_1 \|X\|_* = \{\lambda_1(U_1 V_1^T + W)|U_1^T W = 0, WV_1 = 0, \|W\|_2 \leq 1\}. \tag{8.7}$$

Note that the problem in (8.1) is equivalent to the following *convex* optimization problem,

$$\min_{X,E} \frac{1}{2} \|Z - X - E\|_F^2 + \lambda_1 \|X\|_* + \lambda_2 \|E\|_1.$$

The first-order optimal condition is satisfied by the obtained solution as shown in (8.7) and (8.4). Since the optimization problem is convex, we can conclude that the solution is also global optimal.

$\square$

# 9 Technical Lemmas

**Lemma 3** [Corollary of Theorem 4.1 from Bonnans & Shapiro (1998) ] Let $f : \mathbb{R}^p \times \mathbb{R}^q \to \mathbb{R}$. Suppose that for all $\mathbf{x} \in \mathbb{R}^p$ the function $f(\mathbf{x}, \cdot)$ is differentiable, and that $f$ and $\nabla_{\mathbf{u}} f(\mathbf{x}, \mathbf{u})$ the derivative of $f(\mathbf{x}, \cdot)$ are continuous on $\mathbb{R}^q \to \mathbb{R}$. Let $\nu(\mathbf{u})$ be the optimal value function $\nu(\mathbf{u}) = \min_{\mathbf{x} \in C} f(\mathbf{x}, \mathbf{u})$, where $C$ is a compact subset of $\mathbb{R}^p$. Then $\nu(\mathbf{u})$ is directionally differentiable. Furthermore, if for $\mathbf{u}_0 \in \mathbb{R}^q$, $f(\cdot, \mathbf{u}_0)$ has a unique minimizer $\mathbf{x}_0$ then $\nu(\mathbf{u})$ is differentiable in $\mathbf{u}_0$ and $\nabla_{\mathbf{u}} \nu(\mathbf{u}_0) = \nabla_{\mathbf{u}} f(\mathbf{x}_0, \mathbf{u}_0)$.

**Definition 1** (Quasi-martingale, Fisk 1965). *Let $(\Omega, \mathcal{F}, P)$ be a measurable probability space. A stochastic process $\{X(t), F(t); t \in T\}$ is called a quasi-martingale if there exists a martingale process $\{X_1(t), F(t); t \in T\}$ and a process $\{X_2(t), F(t); t \in T\}$ with almost everywhere sample function of bounded variation on $T$ such that*

$$P([X(t) = X_1(t) + X_2(t); t \in T]) = 1,$$

*where $[\cdot]$ denotes the subset of $\Omega$ for which the argument is true.*

A quasi-martingale process can be decomposed as martingale process plus process of bounded variation.

**Lemma 4** [Sufficient condition of convergence for a stochastic process, Fisk 1965] Let $(\Omega, \mathcal{F}, P)$ be a measurable probability space, $u_t$, for $t \geq 0$, be the realization of a stochastic process and $\mathcal{F}_t$ be the filtration determined by the past information at time $t$. Let

$$\delta_t = \begin{cases} 1 & \text{if } \mathbb{E}[u_{t+1} - u_t | \mathcal{F}_t] > 0, \\ 0 & \text{otherwise.} \end{cases}$$

If for all $t$, $u_t \geq 0$ and $\sum_{t=1}^{\infty} \mathbb{E}[\delta_t(u_{t+1} - u_t)] < \infty$, then $u_t$ is a quasi-martingale and converges almost surely. Moreover,

$$\sum_{t=1}^{\infty} |\mathbb{E}[u_{t+1} - u_t | \mathcal{F}_t]| < +\infty \ \ a.s.$$

**Lemma 5** [Van der Vaart 2000] Let $F = f_\theta : \chi \to \mathbb{R}, \theta \in \Theta$ be a set of measurable functions indexed by a bounded subset $\Theta$ of $\mathbb{R}^d$. Suppose that there exists a constant $K$ such that

$$|f_{\theta_1}(x) - f_{\theta_2}(x)| \leq K \|\theta_1 - \theta_2\|_2,$$

for every $\theta_1$ and $\theta_2$ in $\Theta$ and $x$ in $\chi$. Then, $F$ is P-Donsker. For any $f$ in $F$, let us define $\mathbb{P}_n f$, $\mathbb{P} f$ and $\mathbb{G}_n f$ as

$$\mathbb{P}_n f = \frac{1}{n} \sum_{i=1}^{n} f(X_i), \quad \mathbb{P} f = \mathbb{E}_X[f(X)],$$

$$\mathbb{G}_n f = \sqrt{n}(\mathbb{P}_n f - \mathbb{P} f).$$

Let us also suppose that for all $f$, $\mathbb{P} f^2 < \delta^2$ and $\|f\|_\infty < M$ and that the random elements $X_1, X_2, \ldots$ are Borel-measurable. Then, we have

$$\mathbb{E}_P \|\mathbb{G}_n\|_F = O(1),$$

where $\|\mathbb{G}_n\|_F = \sup_{f \in F} |\mathbb{G}_n f|$.

**Lemma 6** [Positive converging sums, Bertsekas 1999] Let $a_n, b_n$ be two real sequences such that for all $n$, $a_n \geq 0$, $b_n \geq 0$, $\sum_{n=1}^{\infty} a_n = \infty$, $\sum_{n=1}^{\infty} a_n b_n < \infty$, $\exists K > 0$ s.t. $|b_{n+1} - b_n| < K a_n$. Then, $\lim_{n \to \infty} b_n = 0$.

## 10   Empirical Evaluation for Robust PCA

To investigate the robustness of the OR-PCA and GRASTA in details. We plot the performance curve of these two methods under setting where the number of samples $n = 1000$, the ambient dimension $m = 200$, the intrinsic rank $r = \{10, 18, 26, 34, 42, 50\}$, and the corruption fraction $\rho_s$ varies from 0.01 to 0.49. The performance is plotted in Figure 1. From the results, we can make following observations: (I) Under the relatively small corruption fraction (*e.g.*, $\rho_s < 0.17$ when $r = 34$), GRASTA performs a little better than OR-PCA. The reason is that the sample size is relatively small and may be not enough for OR-PCA converging. (II) Under large corruption fraction, the OR-PCA is much more robust than GRASTA. GRASTA will break down rapidly along with the corruption increasing. While even for $50\%$ corruption, OR-PCA still offers around 0.5 E.V. value. The experimental evaluation results clearly demonstrate the robustness advantage of OR-PCA over GRASTA.

Figure 1: The performance comparison of the OR-PCA and GRASTA under different values of the intrinsic rank and the corruption fraction $\rho_s$.

## 11   Robust Subspace Tracking

Besides identifying a static subspace, OR-PCA is also able to solve the subspace tracking problem Crammer (2006), where the underlying subspace of the observations is time variant, due to OR-PCA updating the subspace estimation dynamically. In practice, several important problems can be abstracted as the subspace tracking problem, such as video surveillance with moving cameras, network monitoring. In this subsection, we investigate the performance of online RPCA for tracking the dynamic subspace which is rotated gradually, and compare its performance with the batch RPCA method. In particular, we rotate an initial subspace basis $U_0 \in \mathbb{R}^{p \times r}$ along with the time instance $t$ through $U_t = e^{\delta t B} U_0$. Here $B$ is a randomly generated skew-symmetric matrix[1] and $\delta$ is a parameter to control the rotation degree at each time instant. We generate one observed sample based on each basis $U_t$, following the data generation scheme as in the above subsection. The set of generated corrupted samples $\{\mathbf{z}_1, \ldots, \mathbf{z}_n\}$ forms the streaming samples, which are from different subspaces. In this case, the batch RPCA method will fail since it treats all the samples as from the same subspace. However, the proposed OR-PCA continuously updates the subspace estimation according to each revealed sample. Therefore, it is able to track the rotating subspace. In the simulations, we generate $n = 1,000$ samples with $p = 400$, under the setting of the rank $r = 40$ and outlier fraction $\rho_s = 0.1$. We implement the Principal Component Pursuit over all the $1,000$ samples as the baseline, *i.e.*, batch RPCA. Both the OR-PCA and the batch RPCA are implemented

10 times under each each setting and the average E.V. and the variance are reported. Smaller $\delta$ means the subspaces change more slowly. The subspace recovery performance is also measured by E.V. as aforementioned. Note that the groundtruth subspace is different at different time instance.

We first compare the subspace tracking performance of OR-PCA with batch RPCA under the setting of $\delta = 1$, namely the subspace changes relatively fast. Their performance curves against the number of samples are plotted in Figure 2. From the results, we can make the following observations: (1) For the first $40$ samples, the performance of OR-PCA increases very fast, from the initial E.V. of $0.1$ to $0.5$. This is because the initial samples are from similar subspace and can help improve the subspace estimation well. Then OR-PCA enters a stable state of tracking the subspace and its performance converges to about $0.55$. (2) For the batch RPCA method, because the subspace is changing, its performance is not stable. For the first $400$ samples, the performance keeps increasing. But after that, its performance breaks down soon. (3) Generally speaking, OR-PCA outperforms the batch RPCA with a performance margin of at least $10\%$ under the current setting.

Figure 2: (a) The performance comparison of the online RPCA (blue line) on rotating subspaces with the batch RPCA (red lines) method. The underlying subspace is rotated with the parameter $\delta = 1$. (b) The performance of the OR-PCA on tracking rotating subspaces under different values of the changing speed parameter $\delta$.

Intuitively, the performance of the subspace tracking methods is affected by the speed of the subspace changing. To investigate the ability of OR-PCA to track subspace with different changing speed, we conduct the experiments under the different values of the parameter $\delta = \{0.001, 0.01, 0.1, 1, 10\}$. The performance curves are shown in Figure 2(b). From the results, we can observe that the more slowly subspace rotates, the better OR-PCA performs for tracking. When the changing speed increases, *e.g.,* $\delta = 10$, the performance will drop after achieving the best performance. And finally OR-PCA converges to a relatively low performance.

## Footnotes

[1]We use the MATLAB built-in function *skewdec* to generate the matrix $B$, and then normalize its elements to less than 1, *i.e.*, $B = B/\|B\|_\infty$.