[Reviews · NeurIPS 2013]

Submitted by Assigned_Reviewer_5

The paper presents a iterative algorithm to a robust principal component matrix factorization. The data is modeled as a sum of a low rank matrix approximation and a sparse noise matrix. Constraining the norms of the row and column factors of the former part of the sum allows to implement the nuclear norm minimization of the batch robust pca algorithm in an online stochastic gradient descent fashion.

The approach taken by the authors resembles very much the approach taken by Marial et al 2009 (ICML) and 2010 (JMLR), only that the objective is slightly different (Robust PCA was not dealt with in the JMLR version). One difference between the JMLR and the ICML version is that a standard stochastic gradient version of the objective function did not perform as well as the proposed online dictionary learning approach (in which the statistics of the data are accumulated in the matrices A and B) in the ICML version, but in the JMLR version, a standard stochastic gradient implementation with appropriately chosen learning rate seemed to perform ok. As storing and updating the matrices A and B can be costly, it would be interesting to see whether a standard first order or quasi-newton stochastic gradient approach for optimizing eq. (2) performs as well as the solution proposed in algorithms 1 and 2.

Quality
The manuscript is very well written.

Clarity
All parts of the manuscript appear to be very accessible. Problem formulation and motivation for the method are clear, objective function and the algorithm are comprehensible. Simulations are sensible and well done.

Originality
The approach proposed by the authors seems very similar to the one proposed by Marial et al in their 2009 and 2010 paper.
The proofs appear to be based on the same techniques, in large parts.
Convergence to a global optimum instead of a local optimum is new.
Using a low-norm factorization instead of a batch version of nuclear norm minimization is fairly well established, see e.g. Rennie and Srebro, 'Fast Maximum Margin Matrix Factorization for Collaborative Prediction', ICML, 2005.

Significance
Iterative methods are becoming more relevant with growing data set sizes. It is important to have scalable versions of simple methods such as PCA and robust versions thereof for large scale data sets. Although the ingredients are well known, to the best of my knowledge, the proposed method is novel and potentially relevant to a large part of the research community.
Summary: The paper presents a iterative algorithm to a robust principal component matrix factorization. The data is modeled as a sum of a low-norm low rank matrix approximation and a sparse noise matrix. The basic algorithm as well as the idea of low-norm low-rank matrix factorizations are not new. However the algorithm as well as the theoretical result could be interesting to a large community.

Submitted by Assigned_Reviewer_6

The paper presents an online algorithm for robust PCA (RPCA) by reformulating the optimization problem using an equivalent formulation of nuclear norm. The writing can be improved.

My main concern was with the reformulation of the robust PCA problem which is at the heart of this paper. Having seen the authors' response and discussing the issue with other reviewers, and digging into some previous literature, I believe the math does go through, i.e. the robust PCA problem in equation (1) is indeed equivalent to the optimization problem in equation (2). However, I strongly recommend writing that part carefully and citing "Rennie and Srebro, Fast Maximum Margin Matrix Factorization for Collaborative Prediction, ICML, 2005" which was also suggested by one of the other reviewers -- the result that the authors need can be found clearly stated in that paper.

Summary: The paper presents an online algorithm for robust PCA (RPCA) by reformulating the optimization problem using an equivalent form of nuclear norm.

Submitted by Assigned_Reviewer_7

The paper provides a novel online algorithm for sparse PCA and contributes some theoretical guarantees on the convergence of the algorithm. These are achieved by rewriting the batch problem as a convex optimisation one using the nuclear norm (Principal component pursuit), and by then showing that the online problem (which is not convex naively) is equivalent to a convex one (in the sense that the solutions of the non convex problem correspond to solutions of an alternative convex one). In general, I found the paper well written and I could follow it despite not being an expert in the field. The paper did feel a bit incremental, but the theoretical guarantees are certainly a good point in favour of its quality. The experimental evaluation also seemed thorough and certainly there is considerable motivation for developing algorithms along these lines, which argues for the significance of the paper.
Summary: A good paper describing a novel online robust version of possibly the most popular data analysis algorithm, PCA.

Submitted by Assigned_Reviewer_8

This paper proposes to solve the Low-rank+Sparse approximation problem in an online fashion. One problem with the Principal Component Pursuit algorithm is in order to get a hold on the nuclear norm of the low rank part, one has to feed the algorithm with a batch of samples. The trick in the paper is to use an variational expression of the nuclear norm by introducing additional variables (basis times coefficient) as a “decoupling”. Then the optimization is divided into two phases:
1. fix the basis, and solve a convex programming
2. fix the coefficient and the sparse part, and then update the basis using block coordinate descent with warm start, which is very similar to the online dictionary learning proposed by Mairal et. al.
The authors are also able to prove an asymptotic global convergence guarantee of the algorithm.

To me this is definitely a great paper with both nice theoretical analysis and extensive empirical evaluation. The idea to introduce the variational expression and decouple the problem into the online formulation is very clever. Below are some suggestions and questions:
1. line 282: it looks to me the term \lambda_1 L should not appear on the right hand side.
2. line 267: the condition “if e^\star_{\Lambda}\neq 0” is a little bit confusing to me: According to the definition of \Lambda in line 271, e^\star_{\Lambda} should not equal 0. I don’t understand why you put the “if” there.
3. Theorem 1 and 6 prove that if the solution L produced by the algorithm is full rank, then it is the optimal asymptotically. On the other hand, L is an auxiliary variable matrix that does not appear in the original objective of PCP. Is there any guarantee when the produced L is full rank?

Below are some possible typos in the supplementary:
1. Proof of Lemma 3 in the appendix should be Proof of Lemma 2.
2. Some expressions \tilde{f}(r,e,z,L) in the Proof of Lemma 2 miss the “tilde”.
3. Again the term \lambda_1 L should not appear in one equation in the proof of Lemma 2. Correct me if I am wrong.
4. (8.1) the minimization is with respect to L, R and E
5. (8.1) the middle term is missing \lambda_1
Summary: The idea to introduce the variational expression and decouple the problem into the online formulation is clever. If nobody has done that before, this paper should be accepted.
Author Feedback

Author rebuttal: We thank all reviewers for their constructive suggestions and insightful comments.

The main critique we receive is about the correctness of Equation (2), from reviewer 6. Indeed the reviewer has some misunderstanding, and we hereby clarify the rationale of our reformulation from Eq.(1) to Eq.(2).

The first issue is whether we can fix the size of L and R. For a low rank matrix X whose rank is upper bounded by r, it can be factorized as X = LR’ in its nuclear reformulation, with size of L fixed as m by r and R as n by r. Namely, \|X\|_* = \min_{L \in R^{m x r}, R \in R^{n x r}, X=LR’} 1/2*(\|L\|_F^2 + \|R\|_F^2). This equivalent reformulation, known as low-rank factorization, is developed in (Burer and Monteiro, Math. Progam. 2003) and has been adopted in previous works (e.g., see Sec. 5.3 in B. Recht, SIAM review 2010 and proof therein). The comment from Reviewer 6 that the size of factors L and R cannot be fixed is not correct.

Indeed, when we wrote the reformulation in line 147, we meant the minimization is over L of fixed size n by r and R of m by r, following (Recht, 2010). The size constraint is stated in line 149. Reviewer 6 may misunderstand that line 147 is over all possible L and R. We will put this size constraint explicitly in line 147 and Eq. (2), to avoid any possible confusion.

The second issue is the correctness of (2). Reviewer 6 argues that after replacing the variable X with LR’, the constraint X =LR’ cannot be dropped. This is incorrect. Here we give a step-by-step derivation from (1) to (2) to clarify it:
1. The original problem is \min_X \|Z-X-E\|_F+\|X\|_* +\|E\|_1, as in (1);

2. Recall \|X\|_* = min_{L \in R^{m x r}, R \in R^{n x r}} 1/2*(\|L\|_F^2 + \|R\|_F^2), s.t. X=LR’;

3. Substituting the above to (1) yields \min_{X, L, R}} \|Z-X-E\|_2 + (\|L\|^2_F + \|R\|^2_F)/2 +\|E\|_1 , s.t. X=LR’;

4. Substituting the constraint X=LR’ into the objective yields \min_{L, R}} \|Z-LR’-E\|_2 + (\|L\|^2_F + \|R\|^2_F)/2 +\|E\|_1. This is exactly what is given in (2).

Based on the above explanation, we have to say that Reviewer 6 misunderstands the critical formulation, and gives improper assessment on this paper based on that.

Followings are the point-to-point replies to the reviewers’ other comments.

To reviewer 5
Q1. … it would be interesting to apply a standard stochastic gradient approach for optimizing (2)

A1. We agree with that standard stochastic gradient (SG) may further enhance the computational efficiency. Using current method is motivated by its global convergence guarantee, which may not hold for the standard SG method. The investigation of the SG will be left for future research.

Q2. Using a low-rank factorization is fairly well established, see e.g. Rennie and Srebro.

A2. Thanks for pointing out the related work, which we will cite and discuss.

To Reviewer 6

Q1. The computational complexity is O(pr^2). It is not clear how. What is the computational complexity of step 2 in Alg. 1? How does step 2 scale with p and r?

A1. In step 2, we apply gradient descent, whose complexity is O(r^2+pr+r^3). Here O(r^3) is for computing L^TL. Thus, step 2 scales linear in p and cubic in r. The complexity of step 3 is O(r^2+pr). For step 4 (i.e. Alg.2), the cost is O(pr^2) (updating each column of L requires O(pr) and there are r columns in total). Thus the total complexity is O(r^2+pr+r^3+pr^2). Since p >> r, the overall complexity is O(pr^2). We will provide the above details in the revision.

Q2. Compared with batch PCP, the speed-up brought by OR-PCA is p/r^2, considering the iteration needed in the optimization.

A2. If the iteration complexity of batch PCP is O(1), we agree with that the speedup brought by OR-PCA is p/r^2. Note that when comparing OR-PCA with batch PCP, we are more concerned about the *memory cost*. For batch PCP, all data must be loaded into memory. In contrast, OR-PCA processes the data sequentially and the memory cost is independent of dataset size. This is important for big data.

Q3. The comparison in terms of the runtime can be misleading as there may be many factors affecting it …what one should plot is how much runtime was needed to reach a certain objective.

A3. In the simulations, we have controlled all the factors to ensure the comparison is fair, e.g., the baseline was implemented using the optimized code from the authors, and we used the same initialization for the methods.

Reviewer may not notice that we also provide the final performance of the compared methods in Table 1. It shows that OR-PCA costs less time for convergence and achieves better final performance than baseline. For example, from the first column, OR-PCA takes 13 seconds to achieve E.V. = 0.99 but baseline costs 23 seconds for E.V. = 0.54. We also give the average and variance of the results from 5 repetitions to show OR-PCA consistently outperforms baselines.

Q4. There are faster and more efficient PCA algorithms …
A4. This work focuses on developing online *Robust* PCA. The non-robustness of (online) PCA is independent of used optimization method. Thus, implementing the basic online PCA method is enough for comparing robustness.

To reviewer 7
We thank your positive comments on this paper very much.

To reviewer 9

Thanks a lot for pointing out the typos in the paper and supplementary material. We will revise them thoroughly.

Q1. Is there any guarantee when the produced L is full rank?

A1. We need to pre-define the size of matrix L. If the number of L’s columns is equal to the true rank of the underlying subspace, the produced L will be full rank.

Q2. line 267, the condition if e^\star_{\lambda} \neq 0 is confusing.

A2. Yes, the “if” is not necessary given the definition of \lambda.

Q3. line 282: it looks to me the term \lambda_1 L should not appear on the right hand side.
A3. Yes, this is a typo. Also in the proof of Lemma 2, \lambda L should not appear. Thanks for pointing out this.